

# Differential relieving effects of shikonin and its derivatives on inflammation and mucosal barrier damage caused by ulcerative colitis

Hongwei Han[1,2,*], Wenxue Sun[1,2,*], Lu Feng[1,2], Zhongling Wen[1,2], Minkai Yang[1,2], Yingying Ma[1,2], Jiangyan Fu[1,2], Xiaopeng Ma[1,2], Xinhong Xu[1,2], Zhaoyue Wang[1,2], Tongming Yin[2], Xiao-Ming Wang[1,2], Gui-Hua Lu[1,3], Jin-Liang Qi[1,2], Hongyan Lin[1,2] and Yonghua Yang[1,2]

[1] State Key Laboratory of Pharmaceutical Biotechnology, Institute of Plant Molecular Biology, School of Life Sciences, Nanjing University, Nanjing, China
[2] Co-Innovation Center for Sustainable Forestry in Southern China, MOE Key Laboratory of Forest Genetics and Biotechnology, Nanjing Forestry University, Nanjing, China
[3] School of Life Sciences, Huaiyin Normal University, Huaian, China
[*] These authors contributed equally to this work.

Corresponding authors
Hongyan Lin, linhy@nju.edu.cn
Yonghua Yang, yangyh@nju.edu.cn

## ABSTRACT

**Background.** Ulcerative colitis (UC) is one of the most challenging human diseases. Natural shikonin (SK) and its derivatives (with have higher accumulation) isolated from the root of *Lithospermum erythrorhizon* have numerous beneficial effects, such as wound healing and anti-inflammatory activities. Some researchers have reported that hydroxynaphthoquinone mixture (HM) and SK attenuate the acute UC induced by dextran sulfate sodium (DSS). However, no existing study has systemically investigated the effectiveness of SK and other hydroxynaphthoquinone natural derivative monomers on UC.

**Methods.** In this study, mice were treated with SK and its derivatives (25 mg/kg) and mesalazine (200 mg/kg) after DSS administration daily for one week. Disease progression was monitored daily by observing the changes in clinical signs and body weight.

**Results.** Intragastric administration natural single naphthoquinone attenuated the malignant symptoms induced by DSS. SK or its derivatives remarkably suppressed the serum levels of pro-inflammatory cytokines while increasing the inflammatory cytokine interleukin (IL)-10 . Additionally, both SK and alkanin restrained the activities of cyclooxygenase-2 (COX-2), myeloperoxidase (MPO) and inducible nitric oxide synthase (iNOS) in serum and colonic tissues. SK and its derivatives inhibited the activation of nucleotide binding oligomerization domain-like receptors (NLRP3) inflammasome and NF-κB signaling pathway, thereby relieving the DSS-induced disruption of epithelial tight junction (TJ) in colonic tissues.

**Conclusions.** Our findings shed more lights on the pharmacological efficacy of SK and its derivatives in UC against inflammation and mucosal barrier damage.

## INTRODUCTION

Inflammatory bowel disease (IBD), which encompasses Crohn's disease (CD) and ulcerative colitis (UC), is a kind of refractory and relapsing gastrointestinal inflammatory disease. IBD has not only affected the living quality of patients but also increased the risk of colon cancer if not treated in a timely manner (*Ungaro et al., 2017*; *Stone, Mayberry & Baker, 2003*). An increasing amount of evidence indicates that the incidence of colorectal cancer in IBD patients is generally higher than that in the general population (*Terzic et al., 2010*). Currently, traditional treatment strategies for IBD can be classified as three categories, namely, aminosalicylic acid agents, glucocorticoids, and immunosuppressive agents (*Huang et al., 2009*). Typically, 5-aminosalicylic acid remains the preferred drug in treating IBD, which can directly treat enteritis through inhibiting inflammatory reaction in the intestinal tract and exerting antibacterial function. Nonetheless, this drug will be quickly absorbed by the small intestine if taken orally, leading to acute and chronic renal injury. Moreover, the drug cannot reach the inflammatory site, which can hardly meet the treatment goal (*Munk et al., 2004*). In addition, glucocorticoid is likely to develop dependence after long-term application, and disease recurrence is common after drug withdrawal (*Katz, 2004*). By contrast, immunosuppressant is suitable for patients with chronic enteritis, which can inhibit lymphocyte proliferation, to achieve the therapeutic effect; nonetheless, it is associated with strong toxic side effects (*Schroder & Stein, 2003*). The emergence of anti-tumor necrosis factor therapy has made a major advance in the treatment of patients with IBD for the past two decades (*Khan et al., 2019*). However, despite the fact that this therapy is useful to over half of IBD patients, a substantial proportion of patients are primary nonresponders or lose response with long-term use (*Hanauer et al., 2002*; *Ma et al., 2014*). Moreover, this therapy causes some disturbing safety issues encompassing the increased risk of infections and malignant tumor in some groups (*Lichtenstein et al., 2012*). These factors are the reason for searching other safe, mild, durable, and effective drugs to cure IBD.

The disease characteristics of IBD and shortcomings of current treatments forced us to turn to the traditional Chinese medicine (TCM). *Lithospermum erythrorhizon* (*L. erythrorhizon*) is a commonly used TCM in the Chinese pharmacopoeia of the People's Republic of China and is typically abundant in shikonin (SK) derivatives, which have anti-tumor, anti-virus, and anti-inflammatory activities (*Chen et al., 2019*; *Yoshida et al., 2017*). For example, *Fan et al. (2013)* reported that the crude extract of *Arnebia euchroma* is effective on rats with experimental colitis. Dozens of species of SK are available in the Boraginaceae plants, among which the representative ones are SK, β, β-dimethylacryl-SK (β,β-dimethylacryl-SK), acetyl-SK (acetyl-SK), 5,8-dihydroxy-1, 4-naphthoquinone (naphthoquinone), and alkanin (AK, the enantiomeric of SK) with high contents (*Papageorgiou et al., 2006*). They belong to the naphthoquinones that are collectively known as SKs, which typically possess the advantages of bright color, long-lasting effect, and nontoxic side effects. Currently, the efficacy of each individual component of the crude extract has not been studied in detail. The pathological changes in the DSS-induced IBD model are similar to human UC. Thus, it is an ideal model that has

been widely used to study the mechanism of UC and for screening potential drugs (*Farooq et al., 2018*).

In the present work, we successfully established a murine IBD model by treating C57BL/6 mice with DSS and the effectiveness of some natural SKs with high contents in *L. erythrorhizon* were further studied and compared for their efficacy in IBD treatment to explore the potential mechanisms.

## MATERIALS & METHODS

### Chemicals, Regents and antibodies

Chemicals and Regents were described in File S1. All of compounds were characterized by $^{13}$C NMR, $^{1}$H NMR in File S2. All antibodies are described in Table S1. ECL Kit (#34077) was purchased from Thermo Scientific (USA). Haematoxylin-Eosin/HE Staining Kit (# 20170804) was purchased from Solarbio Science & Technology Co. Ltd (Beijing, China).

### Animals

A total of 66 health male C57BL/6 mice weighted 18–20 g (6–8 weeks old) were selected for in vivo experiment. All of them were purchased from Model Animal Research Center of Nanjing University (Nanjing, China), All animal experiments and welfare were treated in strict accordance with the relevant Guidelines for Care and Use of Laboratory Animals of Nanjing University and approved by the Laboratory Animal Ethics Committee of School of Life Sciences, Nanjing University (IACUC-1909002). The mice were group-housed in a laboratory with controlled conditions ($22 \pm 2$ °C and $60\% \pm 5\%$ humidity) under a 12 h light-dark cycle throughout the experiment. Animals were supplied with standard diet and sterile water ad libitum. All efforts were aimed at reducing the number and suffering of experimental animals while meeting the needs of the experiment.

### DSS-induced colitis and design of drug treatment

The mice were randomly divided into eight groups ($n = 6$ per group) after one week of acclimatization, including the model (the dextran sulfate sodium; 3.5% (w/v) DSS), SK (6.25, 12.5, 25 mg/kg), AK (25 mg/kg), naphthoquinone (25 mg/kg), acetyl-SK (25 mg/kg), β, β-dimethylacryl-SK (25 mg/kg), mesalazine (positive drug; 100, 200 mg/kg) and control group, then all the groups mice except for control group were exposed to 3.5% DSS, which was dissolved in drinking water. All drugs were dissolved in olive oil and were perfused 200 μL per mice once daily at a predetermined dose by gavage which started at the same time as the DSS treatment and continued one week. In the control and DSS-induced groups, the mice fed orally with the same amount of olive oil. There was no significant difference in water intake among all groups. Body weight and the disease activity index (DAI) were measured and recorded everyday (*Farooq et al., 2018*). The DAI was calculated according to the standard listed in Table S2. After one week of treatment, the entire mice monocular eyeball in each mouse was removed under lightly anesthetized (carbon dioxide euthanasia) for blood collection. Then, all mice were euthanized by cervical dislocation in the unconscious state. The colonic tissues were quickly and safely removed and flushed with ice-cold PBS for pathological analysis. The same protocol was carried out at least three times in independent experiments.

## Measurement of Cytokines, COX-2, MPO and iNOS in serum

After drug treatment, mice were sacrificed at the end of the experiment. The whole bloods samples were collected from peripheral blood after reperfusion. Afterwards, the blood was allowed to clot by standing at room temperature for 30 min, and later the clot was removed by centrifuging at 2,000 g for 10 min in a refrigerated centrifuge. The resultant supernatant (serum) was then immediately transferred into a clean polypropylene tube. Kits were used to assay the level of various cytokines (IL-6, IL-1β, TNF-α and IL-10), COX-2, MPO and iNOS in the serum of mice according to the product's guidelines (*Yang et al., 2020*).

## Haematoxylin & Eosin (H&E) staining

Small sections of colonic tissue were fixed in 10% buffered formalin and embedded in paraffin. Thereafter, the sections were stained with haematoxylin and eosin (H&E) and the colonic mucosa was histologically evaluated by a pathologist in a blinded fashion, which was widely used as evaluation criterion (*Fan et al., 2013*).

## Immunohistochemical analysis

Immunohistochemistry was performed for Cyclooxygenase-2 (COX-2), NF-κB, NLRP3, the assay was conducted as described in (*Xu et al., 2019*). Briefly, the tissues were incubated with primary antibodies: anti-NF-$\kappa$B p65 (1:100), anti-COX-2 (1:200) and anti-NLRP3 (1:100) at 4 °C for at least 12 h. Then, the sections were wash with PBS thrice and added with Alexa Fluor 488 labeled anti-mouse secondary antibody (Invitrogen, USA) at 25 °C for 60 min. Signals were developed with Hematoxylin and DAB (Dako, Agilent Technologies, USA). Sections were examined using light microscopy and image analysis software (Image-pro plus 6.0; Media Cybernetics, Inc., Rockville, MD, USA) was used. The positive cumulative optical density (IOD) and the tissue pixel AREA (AREA) of each photo were measured, and thus average optical value (AO) was calculated by using AO=IOD/AREA.

## Western blot analysis

Colonic samples were lysed with RIPA lysis buffer which containing 1% PMSF at 1:100 dilution on ice for 30 min. The insoluble components of cell lysates were removed by centrifugation (10,000 rpm at 4 °C for 10 min) and the harvested protein concentrations were assessed by using the BCA Protein Assay kit. Then the following immunoblot analysis was referred to the study by *Farooq et al. (2018)*. Finally, the immunoblots were quantified by densitometry with Image J Software (National Institutes of Health, Bethesda, Maryland, USA).

## Data and Statistical analysis

Results were presented as mean ± S.E.M. of three independent experiments, with six mice ($n = 6$) per group to ensure their reliability. Statistical comparisons between the treated and untreated groups were performed using Student's $t$-test (two-tailed) using GraphPad PRISM5. Differences with a $P$-value $< 0.05$ were considered statistically significant.

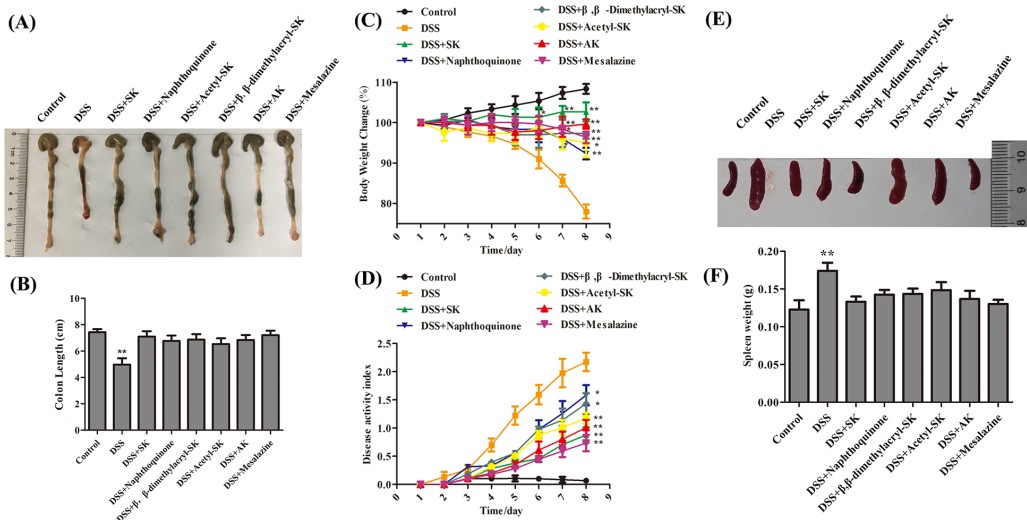

**Figure 1** **SK andits derivatives protect the colon from DSS-induced damage.** (A) Macroscopic observation of colon length. (B) Bar graph showing colon length. ** $p < 0.01$ versus control group. (C) Pattern of daily weight changes. $\star$ $p < 0.05$, ** $p < 0.01$ versus DSS-treated group. (D) Disease activity index of mice in each group. # $p < 0.05$, ## $p < 0.01$ versus DSS-treated group. (E) Macroscopic observation of spleen size. (F) Bar graph showing spleen weight. Data are shown as mean ± S.E.M, $\star$ $p < 0.05$ versus control group ($n = 6$ per group).

# RESULTS

## SK and its derivatives protected the colon from dextran sulfate sodium (DSS)-induced damage in an acute ulcerative colitics mouse model

An acute DSS-induced ulcerative colitis (UC) mouse model was established to investigate the effects of SK and its derivatives on colitis. Mice were given either sterile water or 3.5% DSS dissolved in sterile water in the model. Six other groups received 3.5% DSS and were administered SK (25 mg/kg), naphthoquinone (25 mg/kg), β, β-dimethylacryl-SK (25 mg/kg), acetyl-SK (25 mg/kg), AK (25 mg/kg), and mesalazine (200 mg/kg) orally both at the beginning of the experiment ($n = 6$ mice per group). SK and its derivatives could significantly attenuate the severity level of inflammation at the end of this experiment. DSS group mice suffered a shortening of about 30% compared with normal mice in the case of the colorectum length. SK and its derivatives significantly prevented this shortening (Figs. 1A–1B). Moreover, SK and its derivatives-treated mice showed improvements in body weight and disease activity index, as well as the alleviation of splenomegaly (Figs. 1C–1F). Therefore, SK and its derivatives ameliorated the colon syndromes of DSS-induced damage in mice.

## SK and its derivatives ameliorated the DSS-induced histological damage

The colon tissues of mice in each group were stained with H&E staining to observe the pathological changes. The results (Fig. 2A) indicated that compared with other derivatives group, inflammatory cells infiltration significantly decreased, the number of goblet cells

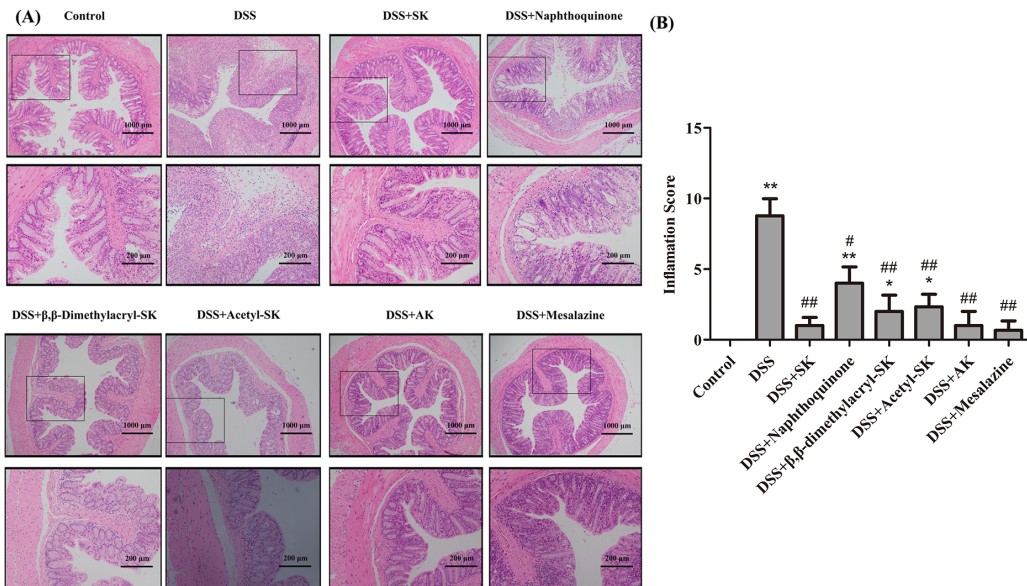

**Figure 2  Colon histology and inflammation score of mice in each group (H&E staining 200 × and 400 ×).** (A) The colon histology sections were stained with hematoxylin and eosin. The colon histology sections of DSS-treated group mice revealed sever pathology with observation of submucosal edema with extensive inflammatory cell infiltration. The frame indicates the region magnified in the bottom panel. (B) Inflammation scores of each group. Data are shown as mean ± S.E.M, *$p < 0.05$, **$p < 0.01$ versus control group; #$p < 0.05$, ##$p < 0.01$ versus DSS-treated group ($n = 6$ per group).

decreased, andcrypt abscesses were distinct reduced in SK and AK-treated group. Grading was performed in a blinded manner by a pathologist and the inflamation score result was showed in Fig. 2B. In brief, the administration of SK and AK markedly relieved the symptoms of DSS-induced IBD in mice.

## SK and its derivatives relieved DSS-induced colonic epithelial TJ disruption in mice

Zonula occludens 1 (ZO-1), vascular cell adhesion molecule-1 (VCAM-1), Occludin and Claudin-1 proteins play an important role in epithelial TJ, which can protect mucosa epithelial cells from breaching by the harmful substances, maintain cellular permeability and integrity, and ensure homeostasis of internal environment. Therefore, the effect of SK and its derivatives on epithelial TJ proteins were further investigated. These results suggested that the expression of ZO-1, VCAM-1, Occludin, and Claudin-1 were remarkably reduced in the DSS group compared with the control group. However, SK and its derivatives at 25 mg/kg can restore the expression of these epithelial TJ proteins (Figs. 3A–3B).

## SK and its derivatives inhibit the activation of NLRP3 inflammasome in colonic tissues

Many studies have revealed that NLRP3 affects several inflammatory disorders to a large extent including colitis (*Lesuis et al., 2012*). Figures 4A–4B shows the significantly up-regulated expression of NLRP3, apoptosis-associated speck-like protein containing a

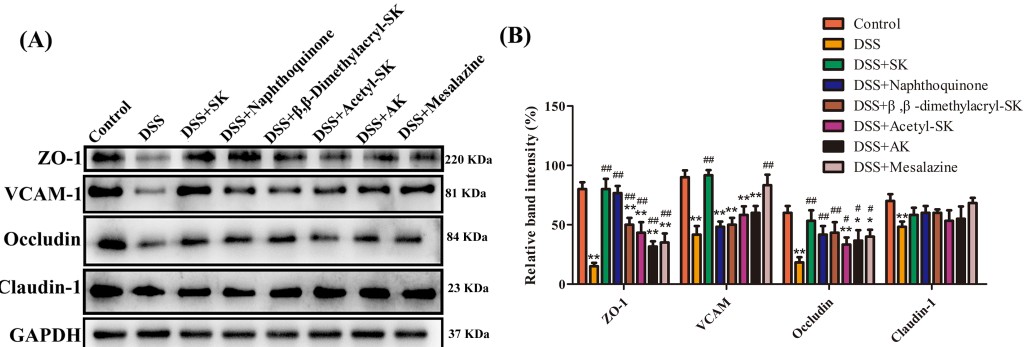

**Figure 3** **SK and its derivatives relieved DSS-induced colonic epithelial tight junction disruption in mice.** (A) Protein expression of ZO-1, VCAM-1, Occludin and Claudin-1 in the colon tissues were analyzed by Western blot. (B) Relative protein expression ratios of were determined by densitometry and normalized to GAPDH. Each point represents the mean ± S.E.M. from three replicates (*$p < 0.05$, **$p < 0.01$ versus control group; #$p < 0.05$, ##$p < 0.01$ versus DSS-treated group).

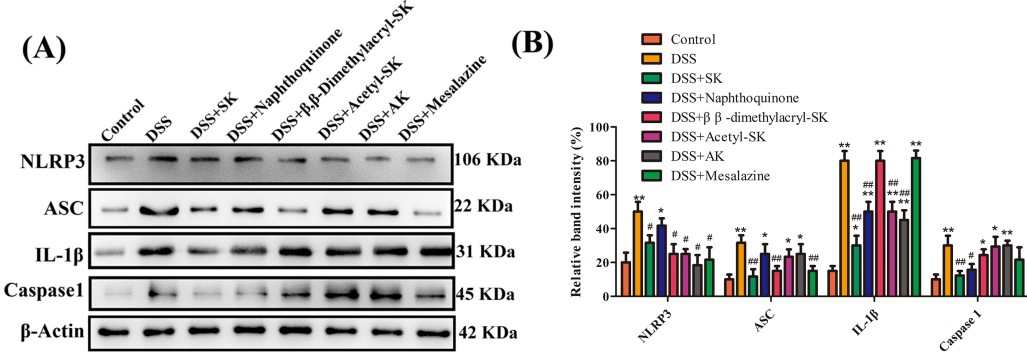

**Figure 4** **SK andits derivatives reduced the level of NLRP3 inflammasome activation in colonic tissues.** (A) Protein levels of NLRP3, ASC, caspase-1, and IL-1β were determined by Western blotting. β- a ctin was used as a control. (B) Relative protein expression ratio was determined by densitometry and normalized to β- a ctin. Each point represents the mean ± S.E.M. from three replicates (*$p < 0.05$, **$p < 0.01$ versus control group; #$p < 0.05$, ##$p < 0.01$ versus DSS-treated group).

CARD (ASC), caspase-1, and IL-1β in DSS group compared with control group. However, changes of caspase-1 and IL-1β were significantly reduced by SK and naphthoquinone. SK- and β, β-Dimethylacryl-SK-treament groups evident reduced the level of ASC protein.

## SK and its derivatives limited the expression of pro-inflammatory cytokines

When the circulating leukocytes are recruited into the colon, pro-inflammatory mediators are released, including pro-inflammatory cytokines such as IL-6, IL-1β and TNF-$\alpha$. The levels of IL-6, IL-1β, TNF-$\alpha$ and IL-10 were detected in the serum from 3.5% DSS-exposed mice given either control diet or SK and its derivatives to determine whether SK and its derivatives attenuated the release of pro-inflammatory cytokines, thereby protecting the intestinal mucosa. Figures 5A–5D showed that treatment with SK and its derivatives at
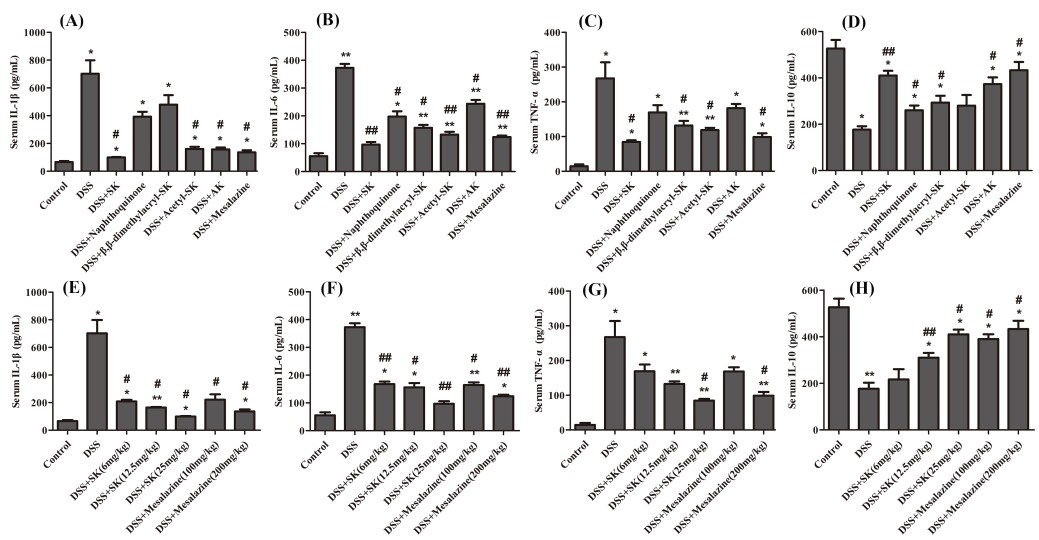

**Figure 5 SK and its derivatives suppressed pro-inflammatory cytokines and promoted anti-inflammatory cytokine in serum.** (A–D) The protein level of cytokines IL-6, IL-1β, TNF-α and IL-10 in serum were determined by ELISA (DSS: 3.5%; SK, Naphthoquinone, β, β-dimethylacryl-SK, Acetyl-SK, AK: 25 mg/kg; Mesalazine: 200 mg/kg). (E–H) Effects of different concentrations of SK on the levels of cytokines IL-6, IL-1β, TNF-α and IL-10. Data are shown as mean ± S.E.M, *$p < 0.05$, **$p < 0.01$ versus control group; # $p < 0.05$, ## $p < 0.01$ versus DSS-treated group ($n = 6$ per group).

25 mg/kg significantly reduced the levels of pro-inflammatory cytokines (IL-6, IL-1β and TNF-α) and increased the level of anti-inflammatory cytokine (IL-10) in the serum. SK showed the best anti-inflammation effect that was similar to mesalazine group at 200 mg/kg. Figures 5E–5H showed that SK and mesalazine (the positive control) reduced the levels of pro-inflammatory cytokines (IL-6, IL-1 β and TNF-α) and increased the level of anti-inflammatory cytokine (IL-10) in a dose-dependent manner, and SK treatment resulted in improved anti-inflammatory effect at a dose of 25 mg/kg. SK and its derivatives may improve the severity of colon damage induced by DSS by inhibiting the expression of pro-inflammatory cytokines.

## SK and its derivatives reduced the activities of MPO, COX-2 and iNOS in serum

MPO is an enzyme that is largely present in neutrophils, and in monocytes and macrophages in small concentrations (*Davies & Hawkins, 2020*) and the high levels of COX-2 (*Chen et al., 2016*) and iNOS (*Pandurangan et al., 2014*) plays a central role in the initiation and propagation of the inflammation. MPO, COX-2 and iNOS concentrations in all the mice groups are shown in Figs. 6A–6C. The MPO activity is an indicator of neutrophil infiltration in inflamed colon tissues. Administration of DSS increased the activity of MPO as compared with normal mice. SK and AK administration at 25 mg/kg markedly reduced the concentrations of MPO, COX-2 and iNOS compared with DSS-induced mice.

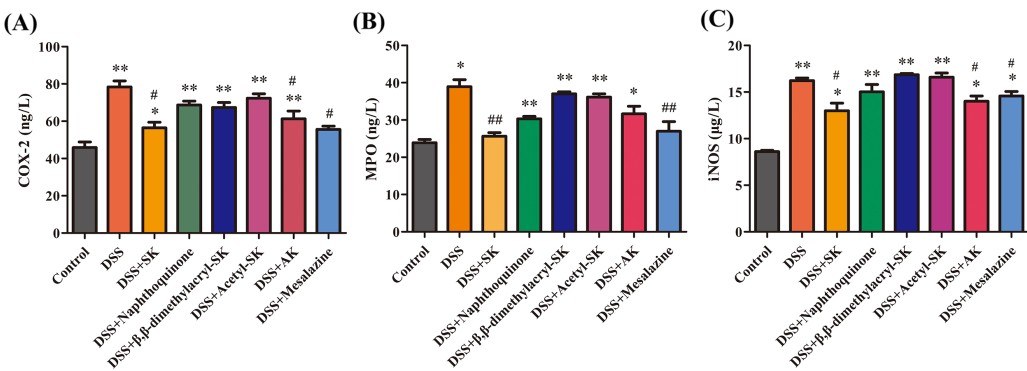

**Figure 6** SK and its derivatives reduced the level of COX-2, MPO and iNOS in the serum of mice. The concentrations of COX-2 (A), MPO (B) and iNOS (C) were measured using an enzyme-linked immunosorbent assay kit. Each point represents the mean ± S.E.M. from three replicates (*$p < 0.05$, **$p < 0.01$ versus control group; # $p < 0.05$, ## $p < 0.01$ versus DSS-treated group).

## SK and its derivatives attenuate the production of COX-2, NF-κB and NLRP3 in colonic tissues

Immunohistochemical expressions of COX-2, NF-κB and NLRP3 in all the groups of mice are shown in Figs. 7A–7C. DSS administration caused the increased production of COX-2, NF-κB and NLRP3 in colonic tissues compared with the control group. Usage of SK and its derivatives reduced the expressions of COX-2, NF-κB and NLRP3 significantly compared with mice induced with DSS.

## SK and its derivatives regulate the COX-2 and iNOS expressions and the activation of NF-κB in colonic tissues

Transcription factors of the NF-κB family play a vital role in regulating genes, which can influence the immune and inflammatory response (*Andujar et al., 2012*). Evidence has verified that NF-κB plays an important role in IBD development. In this disease, NF-κB and STAT-3 activation is observed in inflamed colonic mucosa, leading to the production of COX-2 and iNOS. The phosphorylation level of STAT-3, NF-κB p65 and IκB was detected to evaluate the mechanism of SK. The outcome shown in Figs. 8A–8B suggested that the phosphorylation levels of STAT-3, NF-κB p65 and IκBα were markedly upregulated in DSS-treated group compared with the control group. These increases were markedly attenuated by treatment with DSS plus SK and its derivatives at 25 mg/kg, thereby decreasing the generation of COX-2 and iNOS in the colon homogenates (Figs. 8C–8D).

## DISCUSSION

*L. erythrorhizon* is a kind of TCM used for wound healing in ancient times. Its roots are typically rich with a variety of bioactive substances, including natural naphthoquinones, which represent an ongoing source of therapeutic agents (*Qiu et al., 2018*). Among the naphthoquinones isolated from *L. erythrorhizon*, SK has attracted extensive interest due to its multiple biological activities, such as anti-inflammatory (*Yoshida et al., 2017*; *Yang et al., 2014*), wound healing (*Yao et al., 2019*), anti-microbial (*Andujar et al., 2013*), anti-ulcer

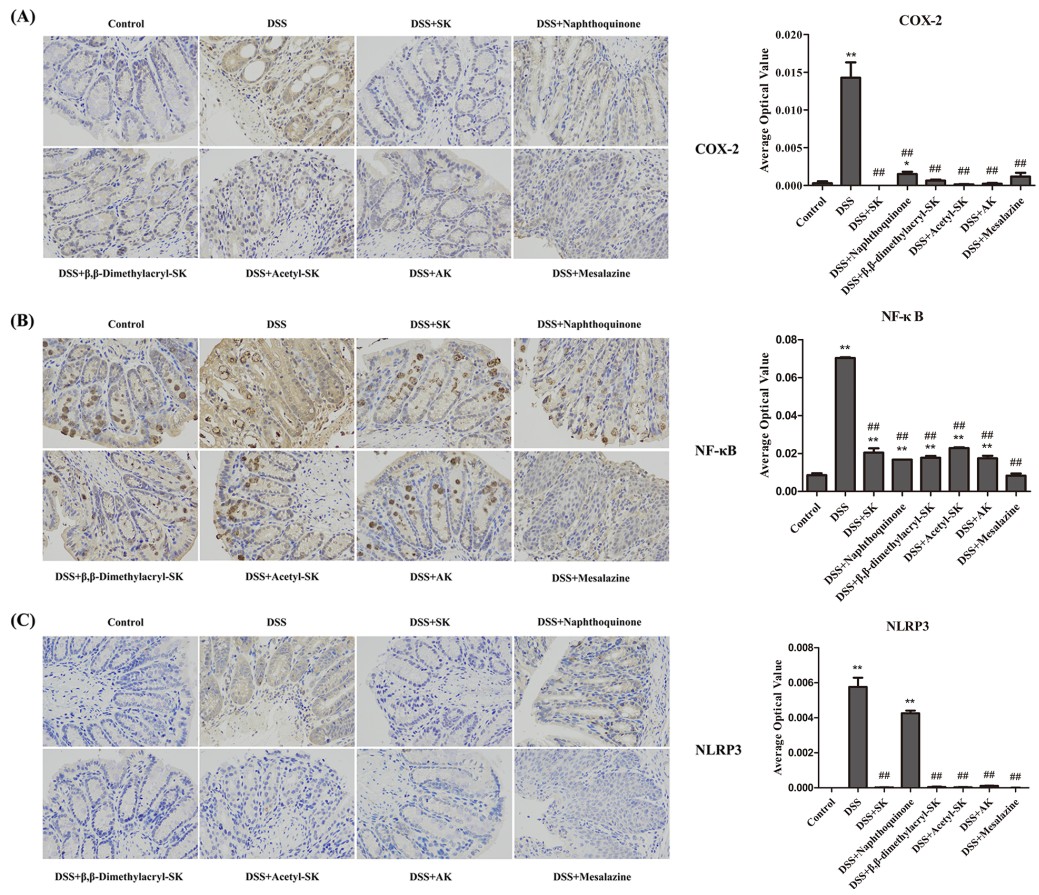

**Figure 7** **SK andits derivatives attenuate the expression of COX-2, NF-κ B and NLRP3.** (A) Immuno-histochemical analysis of COX-2 in each group. (B) Immuno-histochemical analysis of NF-κB in each group. (C) Immuno-histochemical analysis of NLRP3 in each group. The slides were incubated with primary antibody. After the secondary antibody incubation, the slides were developed with DAB and counter-stained with hematoxylin. The images were taken at 400 × magnification. Data are presented as means ± S.E.M. from three replicates (*$p < 0.05$, **$p < 0.01$ versus control group; # $p < 0.05$, ## $p < 0.01$ versus DSS-treated group).

(*Lesuis et al., 2012*), anti-thrombus (*Andrikopoulos et al., 2003*), and anti-cancer activities (*Lin et al., 2018*). Nearly 20 kinds of natural SK derivatives have been identified.

The anti-inflammatory effects of natural naphthoquinones have been extensively investigated. For instance, *Fan et al. (2013)* investigated the effectiveness of hydroxy-naphthoquinone mixture (HM) isolated from *A. euchroma* in rats with experimental colitis induced by 2, 4, 6-tri-nitrobenzene sulfonic acid (TNBS) in 2013. Their results suggested that HM can remarkably attenuate the clinical and histopathological severity of the TNBS-induced colitis in a dose-dependent manner. In addition, HM can also do the following: alleviate loss of body weight, hematochezia, and inflammation; reduce the macroscopic damage score; and improve the histological signs. Moreover, HM can evidently reduce the inflammatory infiltration, ulcer area, and the severity of goblet cell depletion. Notably, HM, which is isolated from *A. euchroma*, contains seven AK

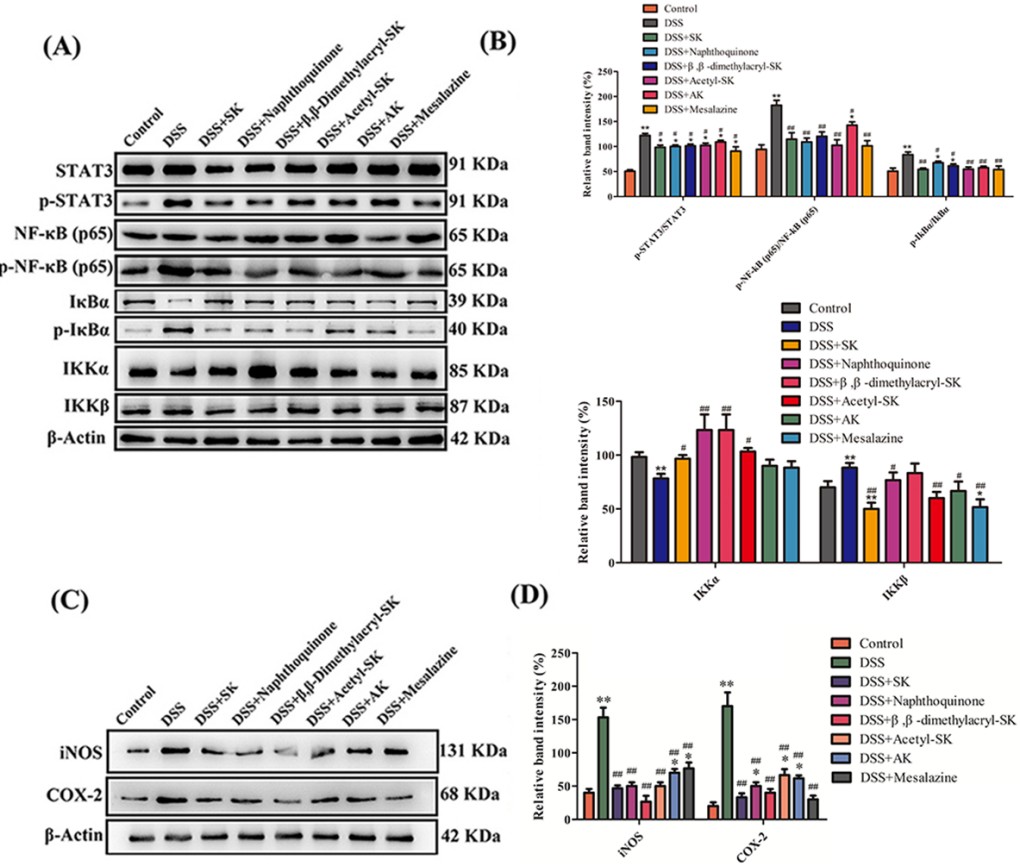

**Figure 8  SK andits derivatives modulated COX-2, iNOS expression, and NF-κB signal pathway in UCmice.** (A) Western blot analysis of COX-2 and iNOS levels in colonic tissues in each group. Data shown are representative of three independent experiments. (B) Relative protein expression ratios of COX-2 and iNOS were determined by densitometry. (C) Western blot analysis of proteins expression level of NF-κB signal pathway. (D) Relative protein expression ratio was determined by densitometry. Each point represents the mean ± S.E.M. from three replicates (*$p < 0.05$, **$p < 0.01$ versus control group; # $p < 0.05$, ## $p < 0.01$ versus DSS-treated group).

derivatives, including AK, deoxyalkannin, acetylalkannin,β, $\beta$-dimethylacrylalkannin, α-methylalkannin, isovalerylalkannin, and β-acetoxyisovalerylalkannin. Isabel et al. reported that the oral administration of SK can attenuate the DSS-induced acute UC by blocking the activation of two major targets, NF-κB and STAT-3 (*Andujar et al., 2012*). However, the effectiveness of the commonly used SK and its natural derivatives on acute colitis has not been systematically investigated yet.

In the present study, five representative natural naphthoquinones, namely, naphthoquinone, SK, acetyl-SK,β, $\beta$-dimethylacryl-SK and AK, were selected to treat DSS-induced colitis in mice. The administration of these five natural naphthoquinones could ameliorate the clinical severity of the wasting disease, prevent the shortening the length of colorectum, decrease the loss of body weight, improve the appearance of feces, and prevent bloody excrements. Such beneficial effects were further proved by histological

evaluation, as evidenced by the markedly reduced severity and extent of inflamed tissue damage and infiltration of inflammatory cells. Typically, the relative expression of pro-inflammatory cytokines, including IL-6, IL-1β and TNF-α in serum in colon tissues, suggested that these five natural naphthoquinones had beneficial effects. MPO is a marker of neutrophil infiltration, which has been observed to be activated in several experimental colitis models, including the DSS-induced colitis. Therefore, MPO is an indicator to quantify intestinal inflammation and evaluate the severity of inflammation. In addition, the pro-inflammatory synthases, including iNOS and COX-2, are inflammatory mediators with vital functions during the pathogenesis of UC (*Itzkowitz, 2006*). The five natural naphthoquinones could suppress MPO, iNOS and COX-2 activities. Such results were consistent with those found in histological examination, in which the inflammatory extent in colonic tissues was less than that in SK- and AK-treated animals and naphthoquinone-, acetyl-SK- and β, β-dimethylacryl-SK-treated mice. Moreover, results of western blot had further verified that SK can relieve the colitis in mice by inhibiting the activation of NF-κB and NLRP3 inflammasome and disruption of epithelial TJ proteins. Collectively, these aforementioned results offer insights into the potential use of SK and its derivatives in treating IBD.

## CONCLUSIONS

Our results suggested that intra-gastric administration of single compound like SK or its derivatives contributed to attenuating the loss of body weight, shortening the colon length, increasing the spleen weight and relieving colonic epithelial tight disruption in mice induced by DSS. Through a number of related pathological experiments, we have explained the underlying mechanism that SK and its derivatives could inhibit the activation of NLRP3 inflammasome and NF-κB signaling pathway. These findings provide new evidence for the potential use of SK and its derivatives to treat the inflammatory bowel disease (IBD).

### Funding

This research was supported by the National Natural Science Foundation of China (NSFC) (U1903201, 21702100, 31771413, 31670298, 21907051), the Program for Changjiang Scholars and Innovative Research Team in University (IRT_14R27), the Natural Science Foundation of Jiangsu Bureau of Science and Technology (BK20191254), and the Fundamental Research Funds for the Central Universities (020814380002, 020814380057). The funders had no role in study design, data collection and analysis, decision to publish, or preparation of the manuscript.

### Grant Disclosures

The following grant information was disclosed by the authors:
The National Natural Science Foundation of China (NSFC): U1903201, 21702100, 31771413, 31670298, 21907051.

The Program for Changjiang Scholars and Innovative Research Team in University:
IRT_14R27.

The Natural Science Foundation of Jiangsu Bureau of Science and Technology:
BK20191254.

The Fundamental Research Funds for the Central Universities: 020814380002,
020814380057.

## Competing Interests

The authors declare there are no competing interests.

## Author Contributions

- Hongwei Han conceived and designed the experiments, performed the experiments, analyzed the data, prepared figures and/or tables, authored or reviewed drafts of the paper, and approved the final draft.
- Wenxue Sun performed the experiments, analyzed the data, prepared figures and/or tables, and approved the final draft.
- Lu Feng and Xiaopeng Ma performed the experiments, prepared figures and/or tables, and approved the final draft.
- Zhongling Wen, Minkai Yang, Yingying Ma, Jiangyan Fu, Xinhong Xu and Zhaoyue Wang analyzed the data, prepared figures and/or tables, and approved the final draft.
- Tongming Yin and Xiao-Ming Wang analyzed the data, authored or reviewed drafts of the paper, and approved the final draft.
- Gui-Hua Lu, Jin-Liang Qi, Hongyan Lin and Yonghua Yang conceived and designed the experiments, analyzed the data, authored or reviewed drafts of the paper, and approved the final draft.

## Animal Ethics

The following information was supplied relating to ethical approvals (i.e., approving body and any reference numbers):

Animal Ethical and Welfare Committee of Nanjing University approved the study (IACUC-1909002).

## Data Availability

Raw data including H&E data, IHC raw data, and Western blot bands, are available in the Supplemental Files.

## Supplemental Information

Supplemental information for this article can be found online at http://dx.doi.org/10.7717/peerj.10675#supplemental-information.

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
