# Peer review of "Differential relieving effects of shikonin and its derivatives on inflammation and mucosal barrier damage caused by ulcerative colitis"

_PeerJ, doi:10.7717/peerj.10675_

## Round 0.1 · original submission · Major Revisions

Please follow the Reviewers' suggestions.

Reviewer 1 ·

Basic reporting

Safe and effective treatments are needed for ulcerative colitis. In this manuscript, the authors have investigated shikonin (SK) and its derivatives as treatments for ulcerative colitis. They have employed a DSS-induced IBD in vivo mouse model and performed comprehensive studies to determine the effects of SK derivatives and understand the underlying mechanisms for the efficacy of shikonin.
-Overall, the article is well-structured and the experiments are well-controlled. The authors have provided sufficient background information in the introduction and shared raw data.
-Rationale for testing COX-2 and iNOS levels needs to be added in the results section in lines 233-239, since that is where COX-2/iNOS is being investigated first.
-Reference is needed to support statements in lines 234-236.
-Abbreviations need to be defined at first mention. For example, ‘DSS’ appears first in the abstract, but is described only in the Methods section. Many other abbreviations are used, but not defined. For example, MPO, iNOS, NLRP3, ASC, ZO-1, VCAM-1.
-The English language should be improved in certain sections of the manuscript, particularly the abstract, lines 136-139, 143-147, 152-154, 159-160, 215. I recommend the authors to use English editing services for this manuscript.

Experimental design

The authors have defined the research question and their research is relevant and within the scope of the journal.
-The ‘Figure titles’ are missing and need to be added. There are no Figure Legends and the symbols * and # used in the Figures to denote statistical significance are not described anywhere in the text.
-Were the cytokine levels determined in the supernatants of cultured colons (line 220) or serum (line 225)?
-In Fig 7, how was the ‘average optical value’ determined? Needs to be added in the Methods section.

Validity of the findings

-In the quantification of Western blot data, what is % relative band intensity?
-Fig 1C: The Y-axis shows % body weight of the original; however, the Y-axis title does not represent that.
-Fig 1F: Is the spleen weight in g or mg?
-Fig 3: The levels of claudin do not look different between DSS and DSS+treatment groups. The conclusion ‘SK and its derivatives at 25 mg/kg can restore the expression of these epithelial TJ proteins’ should be specific since not all TJ protein levels are affected by the SK derivatives.
-Fig 4A: The Western blot results for ASC are not clean.
For IL-1β, the bands for ‘DSS’ and ‘DSS+mesalazine’ appear to be of similar intensity, yet there is a significant difference in the band intensity in Fig 4B. Are the values used to plot Fig 4B correct?
-Fig 8A: IκBα and pIκBα bands are barely visible.
-The authors write that ‘Such results were consistent with those found in histological examination, in which the inflammatory cell infiltration extent in colonic tissues was less than that in SK- and AK-treated animals and naphthoquinone-, acetyl-SK- and β, β-dimethylacryl-SK-treated mice’ (lines 299-302). However, they have not looked at inflammatory cell infiltration specifically. Hence, this statement should be modified.

Reviewer 2 ·

Basic reporting

In this manuscript Han et.al have performed multi-faceted analysis of to evaluate various aspects of IBD with the aim of identifying safe, mild, durable and effective drugs to cure IBD. The authors have shown evidence of pharmacological efficacy of SK on ulcerative colitis against inflammation and mucosal barrier damage. While this is an interesting study, there are some key issues in the manuscript, which if addressed, would make the manuscript stronger.

1. The title and the conclusion suggest that shikonen (SK) and its derivatives are effective against ulcerative colitis. However, majority of studies support role of only SK as an effective pharmacological therapy but not all the other derivatives. The title and conclusion are therefore not entirely in keeping with the results.
2. The language of the title is not very precise. Precise language like “Relieving effects of shikonen on inflammation and mucosal barrier damage caused by ulcerative colitis”, might help the readers understand what the paper is about more clearly.
3. In reference to lines 194 and 195, it is unclear what the authors are trying to convey by saying “SK and its derivatives strengthened the prevention of colon from DSS-induced the histological damage”.
4. Figure 8 A and C: The labeling on top of the figures do not coincide with the lanes of the gel on they are supposed to coincide with. Also, the readers would be able to analyze the figures better if better images for panel IkBa and p-IkBa were provided.
5. In line 218, “When the circulating leukocytes were recruited into the colon, pro-inflammatory mediators...”, “were” (highlighted) should be replaced with “are”.
6. In the background section of the abstract, the abbreviation DSS is used without stating what DSS stands for. Although stated later in the paper, it would be more beneficial for the readers if the abbreviation was explained before using it.

Experimental design

Naphthoquinones are commonly known to cause oxidative DNA damage and increase in reactive oxygen species (ROS) production. As SK is a naphthoquinone, as stated in the paper, it would give the readers more clarity about the by-stander and off-target effects of SK on neighboring cells, if the effect of SK treatment on ROS production and resultant oxidative DNA damage was evaluated.

Validity of the findings

In reference to line 215, it would be more beneficial to the readers if the authors are more precise about reporting which derivatives specifically show an effect on levels of which specific proteins, rather than making a generalized statement like “SK and its derivatives reduce the level of NLRP3 inflammasome activation in colonic tissues “.

---

## Round 0.2 · Major Revisions

There are still concerns to address.

Reviewer 1 ·

Basic reporting

I thank the authors for addressing my concerns in their revised manuscript. There are some more concerns that need to be addressed before publication.

There are still several typos throughout the manuscript and some awkward word usage. For example, in lines 223-224 -However, changes of caspase-1 and IL-1β were significantly “receded” by SK and naphthoquinone.

The added part ‘All animal experiments and welfare were treated in strict accordance with the relevant Guidelines for Care and Use of Laboratory Animals of Nanjing University and approved by the Animal Ethical and Welfare of Nanjing University (IACUC-1909002). All efforts were made to minimize the animals’ suffering and to reduce the number of animals used’ is already present in the original text and therefore, needs to be removed.

Experimental design

No comment

Validity of the findings

Fig 3. Legend is missing and needs to be added.

Fig 5. The ** and ## do not appear to be at the correct place in this figure. There is a clear difference between the control group and the DSS group and yet there is no ** on top of the DSS bars in the graphs. Besides, SK significantly reduces levels of pro-inflammatory cytokines and increases IL-10 and yet, there is no ## on the DSS+SK bars in the graphs.


Authors have described statistics as follows: * p<0.05, ** p<0.01 and # p<0.05, ## p<0.01. The greater the observed difference, higher the statistical significance and lower the p value. So assuming * p<0.05 and # p<0.05 is correct, is ** p<0.01 and ## p<0.01 correct? From the figures, it is clear that the observed difference is greater for ** compared to * and ## compared to #.

Additional comments

In the abstract, SK increased the ‘anti-inflammatory’ cytokine IL-10. It is mentioned as ‘inflammatory’ cytokine IL-10.

---

## Round 0.3 · accepted · Accept

Revisions have been performed properly and the manuscript is ready for publication.

Reviewer 1 ·

Basic reporting

No comment

Experimental design

No comment

Validity of the findings

No comment

Additional comments

I thank the authors for addressing my concerns in the revised manuscript.